# Effect of Vitamin D Supplements on Relapse of Digestive Tract Cancer with Tumor Stromal Immune Response: A Secondary Analysis of the AMATERASU Randomized Clinical Trial

**DOI:** 10.3390/cancers13184708

**Published:** 2021-09-20

**Authors:** Taisuke Akutsu, Kazuki Kanno, Shinya Okada, Hironori Ohdaira, Yutaka Suzuki, Mitsuyoshi Urashima

**Affiliations:** 1Division of Molecular Epidemiology, The Jikei University School of Medicine, Tokyo 105-8461, Japan; t-akutsu@jikei.ac.jp (T.A.); h22ms-kanno@jikei.ac.jp (K.K.);; 2Department of Pathology, International University of Health and Welfare Hospital, Tochigi 329-2763, Japan; shinya1012@iuhw.ac.jp,; 3Department of Surgery, International University of Health and Welfare Hospital, Tochigi 329-2763, Japan; ohdaira@iuhw.ac.jp (H.O.); yutaka@iuhw.ac.jp (Y.S.)

**Keywords:** vitamin D, cancer, survival, CD56, CD3, CD4, CD8, CD45RO, tumor-infiltrating lymphocytes, tumor stroma

## Abstract

**Simple Summary:**

Clinical evidence suggesting the beneficial effects of vitamin D supplementation on survival of patients with cancer has been accumulating. More tumoral immune cells have been shown to be associated with a longer survival time; however, interactions among vitamin D supplementation, intratumoral immune cells, and cancer relapse have not yet been elucidated. The aim was to assess the effects of vitamin D supplementation on relapse in patients with digestive tract cancer showing an immune response in the tumor. A secondary analysis of a randomized clinical trial including 372 patients was performed. In the higher half subgroup of CD56+ immune cells infiltrating in tumor stroma, the relapse ratio was significantly lower in the vitamin D group (7.4%), than in the placebo group (20.5%), whereas in the lower half subgroup, relapse ratio was not different. Vitamin D supplementation may reduce the risk of relapse in patients who already had adequate infiltration of immune cells in their tumor stroma.

**Abstract:**

The aim was to examine whether vitamin D supplementation (2000 IU/day) reduces the risk of relapse in a subgroup of patients with digestive tract cancer, showing a sufficient immune response in tumor stroma by conducting secondary subgroup analyses of the AMATERASU randomized, double-blind, placebo-controlled trial (UMIN000001977). A total of 372 patients were divided into two subgroups stratified by the median density of immune cells infiltrating in tumor stroma into higher and lower halves. In the higher-half subgroup of CD56+ cells, the relapse ratio was significantly lower in the vitamin D group (7.4%) than in the placebo group (20.5%) (subdistribution hazard ratio (SHR), 0.35; 95% confidence interval (CI), 0.15–0.82), but it was equivalent (25.2% vs. 22.7%) in the lower-half subgroup of CD56+ cells (SHR, 1.21; 95% CI, 0.68–2.19) with a significant interaction (*P_interaction_* = 0.02). Although there were no significant differences, the risk of relapse was lower in the vitamin D group than in the placebo group in the higher half of CD45RO+ memory T cells (8.9% vs. 19.2%), and of CD8+ cytotoxic T cells (11.3% vs. 22.5%). In patients with digestive tract cancer, vitamin D supplementation was hypothesized to reduce the risk of relapse in the subgroup of patients who already have an adequate infiltration of immune cells in their tumor stroma.

## 1. Introduction

Clinical evidence suggesting the beneficial effects of vitamin D supplementation on survival of patients with a variety of cancers has been accumulating [1]. Up to now, as mechanisms of anti-tumor effects, vitamin D has been hypothesized to inhibit proliferation, inflammation, and metastasis, as well as to induce apoptosis and differentiation of cancer cells [2]. On the other hand, as mechanisms of anti-infective effects, vitamin D was demonstrated to upregulate both innate and adaptive immunity [3]. In fact, meta-analyses of randomized clinical trials (RCTs) showed that vitamin D supplementation, compared with placebo, prevented the incidence of acute respiratory tract infection [4,5], probably through enhancing innate immunity, such as CD56+ natural killer cells. In addition, more tumoral immune cells have been shown to be associated with longer survival time of patients with various types of cancers, e.g., advanced ovarian cancer and CD3+ pan T cells [6], colorectal cancer and CD45RO+ memory T cells [7], and breast cancer and CD8+ cytotoxic T cells [8]. Moreover, in patients with head and neck cancer, a statistically significantly higher intratumoral and/or stromal infiltration of CD3+ pan T, CD4+ helper T, CD8+ cytotoxic T, CD56+ natural killer cells, and other immune cells was observed in the 25(OH)D-high patients compared with the 25-hydroxyvitamin D (25[OH]D)-low patients, which was associated with longer overall survival [9]. However, interactions among vitamin D supplementation, intratumoral immune cells, and cancer relapse have not yet been elucidated. If vitamin D supplementation inhibits cancer cell growth by enhancing anti-tumor immunity, the risk of relapse may be less in the vitamin D group than in the placebo group in the subgroup of patients with tumor infiltrated by more immune cells, but not by less immune cells. In this secondary analysis of the AMATERASU trial [10], the aim was to examine whether vitamin D supplementation reduces the risk of relapse in the subgroup of patients with digestive tract cancer showing a sufficient immune response in tumor stroma.

## 2. Materials and Methods

### 2.1. Trial Design

This was a secondary subgroup-analysis of parent trial participants, i.e., the AMATERASU trial (UMIN000001977) [10], with high vs. low tumoral stromal infiltration of immune cells. Briefly, the parent trial enrolled 417 patients with digestive tract cancers from the esophagus to the rectum who participated in a randomized, double-blind, placebo-controlled trial to compare the effects of vitamin D3 supplements (2000 IU/day) and placebo on relapse and/or death at an allocation ratio of 3:2 at the International University of Health and Welfare Hospital (Otawara, Tochigi, Japan) between January 2010 and February 2018. The participants were asked to continue the supplements until the end of the trial. The trial protocol was approved by the ethics committee of the International University of Health and Welfare Hospital (ethics approval code: 13-B-263), as well as the Jikei University School of Medicine (Nishi-shimbashi, Tokyo, Japan) (ethics approval code: 21–216 [6094]). Written, informed consent was obtained from each participating patient. This study followed the Consolidated Standards of Reporting Trials (CONSORT).

### 2.2. Participants

Of the 417 patients with digestive tract cancers who were randomly assigned to receive vitamin D supplements (*n* = 251: 60%) or placebo (*n* = 166: 40%), participants with available pathological specimens were eligible for this secondary analysis. Details of the inclusion and exclusion criteria were described in the original report [10].

### 2.3. Outcomes

The primary outcome was relapse, defined as elapsed time from starting the supplement to the date of relapse or censored. The secondary outcome was relapse or all-cause death, as well as all-cause death alone, defined as elapsed time from starting the supplement to the date of outcome occur or censored.

### 2.4. Tissue Microarray

The tissue microarray (TMA) was constructed using pathological specimen uniformly obtained at operation as described in the previous analysis [11], and the results were also used in the present study. This TMA consisted of a 5-mm-diameter core selected by a study pathologist (SO) after observing surgically resected specimen slides with hematoxylin and eosin (HE) staining; the deepest invasion sites, where relapse was expected, such as the resection margin or serosal exposure, were selected by avoiding necrosis.

### 2.5. Immunohistochemistry

Primary antibodies against CD3 (SP-7), CD4 (4B12), CD8 (C8/144B), CD45RO (UCHL1), and CD56 (MRQ-42) (Nichirei Biosciences Inc., Tsukiji, Tokyo, Japan) were used. Immunohistochemistry (IHC) using these antibodies was performed with the Histofine Histostainer 36A, according to the manufacturer’s protocols.

### 2.6. Image Analysis and Scoring

After observing the whole TMA core with a 4× objective lens, the area of the so-called ‘hot spot’ having a large number of positive cells was observed with the 40× objective lens and its image was taken by a study investigator (TA). The images were analyzed using semi-custom image analysis software (e-path Co. Ltd., Fujisawa, Kanagawa, Japan), which is designed to count the number of positive cells by determining detection conditions based on cell staining intensity, size, and shape (length/shortness ratio and roundness) (Figure 1). Each parameter determined for each IHC was uniformly applied to all images in the same series. The other study investigator (KK) distinguished “immune cells in tumor stroma”, defined as a lymphocytic reaction in tumor stroma regions within the tumor mass, and “tumor-infiltrating lymphocytes”, defined as immune cells on top of cancer cells, according to Ogino’s method [12,13]. For each patient, the densities of CD3+, CD4+, CD8+, CD45RO+, and CD56+ immune cells per area in tumor stroma and on top of cancer cells were calculated separately using the image analysis software. All image analysis procedures were performed by the investigators (TA, KK), who were blind to the randomized groups and clinical information, including outcomes, and fixed prior to statistical analyses. The calculated densities for each of CD3+, CD4+, CD8+, CD45RO+, and CD56+ immune cells in tumor stroma and tumor-infiltrating lymphocytes were divided into two subgroups, “higher half” and “lower half”, using the medians as the cut-off values.

### 2.7. Evaluation of Other Covariates

The details of the analysis of histopathological subtypes [14], analysis of p53 protein, vitamin D receptor (VDR), Ki-67 by IHC [11], and serum levels of bioavailable 25(OH)D [15] were described in previous reports (Figure 2).

### 2.8. Statistical Analysis

All patients who underwent randomization and for whom IHC evaluation was available were included in this analysis. Spearman’s rank correlation (rho) was used to quantify the strengths of associations between two continuous variables: rho ≥ 0.9, very strong; 0.9 > rho ≥ 0.7, strong; 0.7 > rho ≥ 0.4, moderate; 0.4 > rho ≥ 0.1, weak; and rho < 0.1, negligible [16]. Relapse and death-related outcomes were assessed by intention to treat analysis. To evaluate the effects of vitamin D supplementation on relapse as the primary outcome, cumulative incidence functions were applied by considering patient deaths due to causes other than cancer relapse as a competing risk; competing risk regression was performed using sub-distribution hazard ratios (SHRs) and 95% confidence intervals (95%CIs) [17]. On the other hand, because the competing risk between relapse and death does not have to be considered, the effects of vitamin D and placebo on the risk of relapse or all-cause death as well all-cause death alone were estimated using Nelson–Aalen cumulative hazard curves for outcomes. A Cox proportional hazards model was used to determine hazard ratios (HRs) and 95%CIs for these outcomes. When the 95%CI did not include 1, the SHR or HR was considered significant. To clarify whether vitamin D supplementation significantly affected these subgroups, the P for interaction was analyzed on the basis of a Cox regression model including three variables, for example, (i) the vitamin D group; (ii) the subgroup of CD3+ higher half; and (iii) both multiplied together as an interaction variable, by two-way interaction tests comparing the subgroups of CD3+ higher half and CD3+ lower half. Values of *p* for interaction with two-sided *p* < 0.05 were considered significant. All data were analyzed using Stata 17.0 (StataCorp LP, College Station, TX, USA).

## 3. Results

### 3.1. Study Population

Of the parent trial, IHC results for CD3+, CD4+, CD8+, CD45RO+, and CD56+ were available for 219 (87%) patients in the vitamin D group and 153 (92%) patients in the placebo group, for a total of 372 patients (89%) (mean age, 66.5 years; male, 66.7%; esophageal cancer, 9.1%; gastric cancer, 42.8%; small bowel cancer, 0.5%; colorectal cancer, 47.6%), due to lack of tissue samples: no cancer tissue availability, a special tissue subtype such as neuroendocrine tumor, or inappropriate samples during the TMA process (Appendix A). Patients analyzed in this study were followed-up for a median 3.5 (interquartile range, 2.2–5.0) years, with maximal follow-up of 7.6 years.

### 3.2. Image Analyses

The densities of CD3+, CD4+, CD8+, CD45RO+, and CD56+ immune cells in tumor stroma were assessed in 372 patients. The scatter plots and Spearman’s rank correlation coefficients for each subset of variables are shown in Appendix A. Similarly, the densities of CD3+, CD4+, CD8+, CD45RO+, and CD56+ tumor-infiltrating immune cells were assessed (Appendix A).

### 3.3. Patients’ Characteristics

A total of 372 patients were divided into two subgroups stratified by the median density of each subset, i.e., CD3+, CD4+, CD8+, CD45RO+, and CD56+ immune cells, in tumor stroma into higher half (*n* = 186) and lower half (*n* = 186). Typical images of CD3+, CD4+, CD8+, CD45RO+, and CD56+ cells in the higher-half subgroups and lower-half subgroups are shown in Appendix A. Patients’ characteristics in the subgroups are shown in Appendix A. There were no differences in serum 25(OH)D levels or bioavailable 25(OH)D levels before vitamin D intervention between the compared subgroups, e.g., between patients with high and low CD56+ cells. On the other hand, of the other variables, Ki67 in particular was significantly higher in the CD3+ higher half than in the lower half (Mann–Whitney test: *p* < 0.0001).

Patients’ characteristics stratified by vitamin D and placebo groups are shown in Appendix A, with a similar distribution of variables to the original trial.

### 3.4. Effects of Interactions between Vitamin D and Immune Cells on Relapse

Considering the competing risk between relapse and death, the effects of vitamin D supplementation on relapse were compared between the higher and lower halves of each immune subset in tumor stroma by SHRs. In the subgroup with the higher half of CD56+ cells infiltrating in tumor stroma, relapse was observed in 8 patients of the vitamin D group (7.4%), which was significantly lower than in the placebo group (16 patients, 20.5%) (SHR, 0.35; 95%CI, 0.15–0.82) (Figure 3A), but equivalent (vitamin D, 28 patients (25.2%) vs. placebo, 17 (22.7%)) in the subgroup with the lower half (SHR, 1.21; 95%CI, 0.68–2.19) (Figure 3B). There was a significant interaction between vitamin D supplementation and the subgroup of CD56+ higher half (*P_interaction_* = 0.02). In other words, vitamin D supplementation effectively reduced the risk of relapse only in the higher half of the CD56+ cells subgroup, but not in the lower half.

Although there were no significant differences, the risk of relapse was lower in the vitamin D group than in the placebo group in subgroups of patients with higher half of CD45RO+ (8.9% vs. 19.2%; SHR, 0.45; 0.20–1.01) (Figure 4A) but the same in the lower half of CD45RO+ (Figure 4B). Similarly, it was lower in the higher half of CD8+ (11.3% vs. 22.5%; SHR, 0.49; 0.24–1.02) (Figure 4C), but the same in the lower half of CD8+ (Figure 4D).

Similar analyses for each subset of tumor-infiltrating lymphocytes were performed and are shown in Appendix A; no significant SHRs or Ps for interaction were obtained in these analyses.

### 3.5. Effects of Interactions between Vitamin D and Immune Cells on Relapse or Death

The effects of vitamin D supplementation on HRs of relapse or death were compared between the higher and lower halves of each immune subset in the tumor stroma. Relapse or death was observed in 12 patients of the vitamin D group (11.1%), which was significantly lower than in the placebo group (19 patients, 24.4%) in the subgroup with the higher half of CD56+ cells infiltrating in tumor stroma (HR, 0.42; 95%CI, 0.20–0.87) (Figure 5A), but equivalent (vitamin D vs. placebo = 32 patients (28.8%) vs. 21 (28.0%)) in the subgroup with the lower half (1.13; 95%CI, 0.65–1.97) (Figure 5B). There was a significant interaction between vitamin D supplementation and the subgroup of CD56+ higher half (*P_interaction_* = 0.03).

In all other immune subsets, the hazards were lower in the vitamin D group than in the placebo group in the higher half of CD45RO+ (Figure 6A) and CD8+ (Figure 6C), but not significantly. In contrast, in patients in the lower half subgroups, the hazard curves between the vitamin D and placebo groups were close to each other (Figure 6B,D). However, interactions between the higher and lower halves were not significant in the subsets of CD45RO+ (*P_interaction_* = 0.19), as well as CD8+ (*P_interaction_* = 0.26).

Similar analyses for each subset of tumor-infiltrating lymphocytes were performed and are shown in Appendix A; no significant HRs or Ps for interaction were obtained in these analyses.

### 3.6. Effects of Interactions between Vitamin D and Immune Cells on Death

The effects of vitamin D supplementation on HRs for death were compared between the higher and lower halves. In either the higher half or the lower half, HRs for death were not significantly different between the vitamin D and placebo groups in each subset of immune cells. However, there was a significant interaction between vitamin D supplementation and the subgroup of CD56+ higher half (*P_interaction_* = 0.03) (Figure 7A,B).

Other interactions were not significant in the subsets of CD3+ (*P_interaction_* = 0.18), CD4+ cells (*P_interaction_* = 0.67), CD8+ (*P_interaction_* = 0.26), and CD45RO+ (*P_interaction_* = 0.19).

Similar analyses for each subset of tumor-infiltrating lymphocytes were performed and are shown in Appendix A; no significant HRs or Ps for interaction were obtained in these analyses.

## 4. Discussion

Vitamin D supplementation, compared with placebo, significantly reduced the risk of relapse by 65% in the subgroup of patients with the higher half of CD56+ in tumor stroma, but it did not change it in the lower half, suggesting a significant interaction between vitamin D supplementation and CD56+ cells. CD56+ cells are considered to be mainly natural killer cells [18], which can kill cancer cells through a cytotoxic reaction as part of innate immunity [19]. Recent meta-analyses supported the notion that high levels of tumor-infiltrating natural killer cells correspond with better survivals in patients with solid tumors [20,21]. On the other hand, dietary vitamin D supplementation can modulate innate immunity by increasing natural killer activity in mice [22]. In view of the results obtained in the present study and previous reports, vitamin D supplementation was hypothesized to suppress relapse by upregulating anti-tumor effects of CD56+ natural killer cells in tumor stroma, at least in part.

On the other hand, other subsets, i.e., CD3+ pan T cells, CD4+ helper T cells, CD8+ cytotoxic T cells, and CD45RO+ memory T cells, are also considered important players in adaptive and cancer immunity. Regarding CD45RO+ and CD8+ cells, although the interactions were not significant, hazard curves of relapse between vitamin D and placebo were widely opened in the higher half, but they were closed in the lower half. The immunoscore is based on the number of two lymphocyte populations, CD45RO+ memory T cells and CD8+ cytotoxic memory T cells, both in the core and in the invasive margin of tumors [23]. In colorectal cancer tissue, CD3+, CD4+, CD8+, and CD45RO+ cells may also be considered to play a critical role in tumor control [24,25]. Thus, vitamin D supplementation may reduce the risk of relapse in the subgroup of patients who already had adequate infiltration of immune cells in the tumor stroma of patients with digestive tract cancer.

In the present study, CD56+ immune cells in tumor stroma, but not tumor-infiltrating lymphocytes, showed significant differences in relapse. Experimental results suggest that vitamin D not only suppresses growth of cancer cells, but also regulates the tumor microenvironment to facilitate tumor repression [26]. High VDR expression in tumor stromal fibroblasts was associated with better overall survival and progression-free survival in patients with colorectal cancer, independently of its expression in carcinoma cells [27]. The CD3+ cell infiltration was higher in patients with high VDR expression in tumor stromal cells. Thus, vitamin D supplementation may suppress the risk of relapse by interacting through VDR expressed in the cells existing within the tumor microenvironment.

In the present study, Ki-67 was significantly higher in the CD3+ higher half than in the lower half. Ki-67 is considered to be an indicator of proliferative activity and thus one of the markers of a poor prognosis in carcinoids [28,29]. However, expression levels of Ki-67 were not always associated with survival of patients with cancer, at least in this cohort. A recent study showed that high tumor cell proliferation assessed by Ki-67 was associated with increased levels of all tumor-infiltrating lymphocyte subsets, including CD3+ pan T cells in breast cancer [30], which is consistent with the results of the present study. Although the exact mechanisms are unknown, cancer cells with high proliferative activity may tend to induce inflammation and attract immune cells.

This secondary analysis has several limitations. First, subgroup analyses of five surface markers of immune cells may increase the probability of false positives due to multiple comparisons. Second, sample sizes were not calculated for the subgroup analyses. There were no significant differences in the hazard curves of CD45RO+, CD8+, and other immune cells between the higher and lower halves. Thus, these results may contain false negatives due to the small sample size of each subgroup. RCTs are the gold standard for evaluating the effects of treatments and are the cornerstone of evidence-based medicine, but they cannot always deal with the heterogeneity of treatment effects [31]. Third, due to the nature of the TMA method, only a small core of the tumor including immune cells in tumor stroma and tumor-infiltrating lymphocytes could be obtained in pathological specimens. Thus, the invasive margin and Crohn’s-like reaction, as well as the peritumoral reaction [12,13], could not be evaluated. Fourth, because the AMATERASU trial was conducted in Japan, the patients were Asian, most esophageal cancers were squamous cell carcinomas, the incidence of gastric cancer was still relatively high, and the bioavailable 25(OH)D could be different from that in other population groups. Thus, the results of the present study are not necessarily generalizable to other populations.

## 5. Conclusions

Vitamin D supplementation may reduce the risk of relapse in the subgroup of patients with digestive tract cancer who already showed adequate infiltration of immune cells in their tumors.

## Figures and Tables

**Figure 1 cancers-13-04708-f001:**
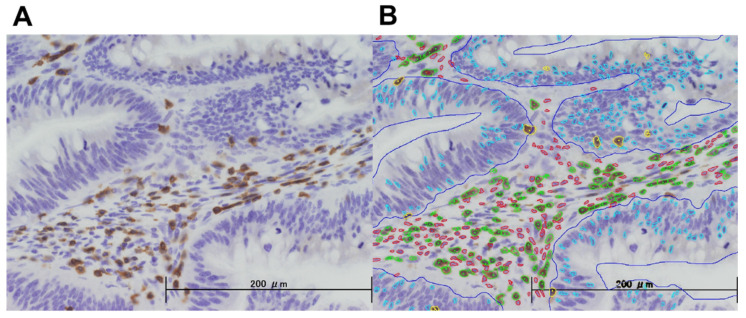
A typical staining pattern of well-differentiated adenocarcinoma in a patient with colon cancer. (**A**) Immunohistochemical staining with anti-CD3 antibody. Brown cells = CD3+ cells. (**B**) The images were analyzed using semi-custom image analysis software. Blue line = boundary between peri-cancerous stromal area and intra-cancerous area within the tumor. Green cells = CD3+ cells infiltrating in tumor stroma. Yellow cells = CD3+ cells on top of the cancer cells. Other cells including those showing red and sky-blue colors were not counted.

**Figure 2 cancers-13-04708-f002:**
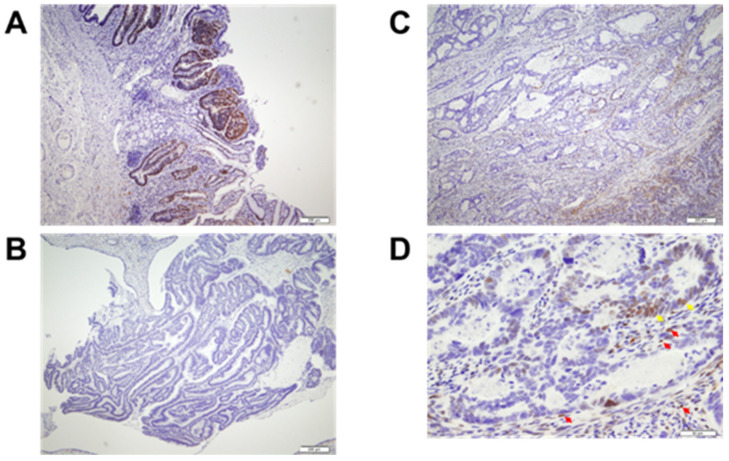
Immunohistochemical staining with anti-vitamin D receptor (VDR) antibody. Brown cells = VDR+ cells. (**A**) Cancer cells, but not stromal cells, are highly expressing VDR (×40). (**B**) Both cancer cells and stromal cells are not expressing VDR (×40). (**C**) Both cancer cells and stromal cells are at least in part expressing VDR (×40). (**D**) A magnified view of a part of panel (×200). Notice that VDR is expressed mainly in the nuclei of both cancer cells (yellow arrows) and stromal cells (red arrows).

**Figure 3 cancers-13-04708-f003:**
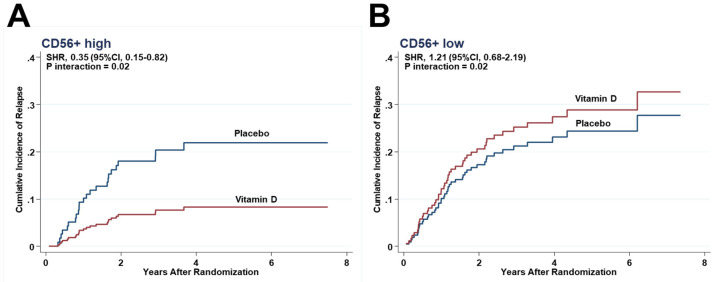
Effect of vitamin D on relapse by competing-risk analysis in the subgroup with the higher half of CD56+ cells (**A**) and the lower half of CD56+ cells (**B**). Cumulative incidence of relapse by competing-risk analysis in the subgroups of (**A**) CD56+ higher half, and (**B**) CD56+ lower half.

**Figure 4 cancers-13-04708-f004:**
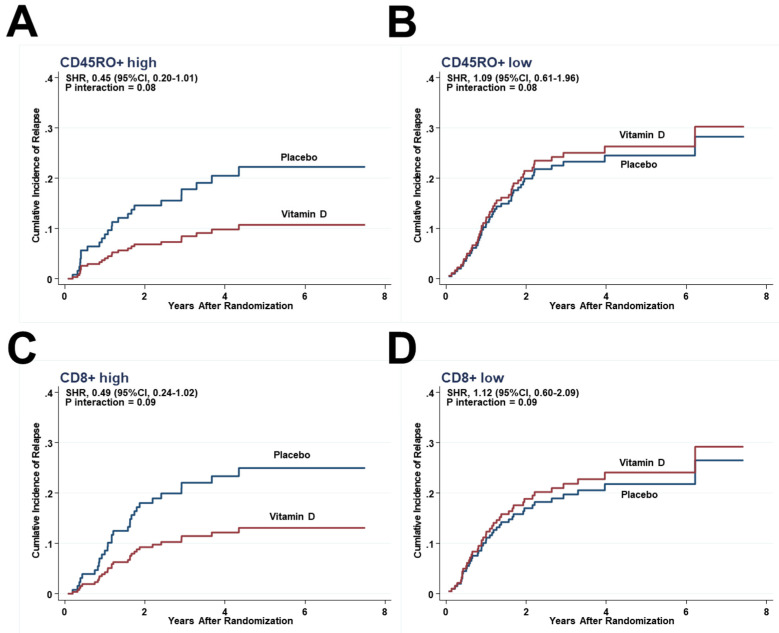
Effect of vitamin D on relapse by competing-risk analysis in the subgroups of (**A**) CD45RO+ higher half, and (**B**) CD45+ lower half, as well as (**C**) CD8+ higher half, and (**D**) CD8+ lower half.

**Figure 5 cancers-13-04708-f005:**
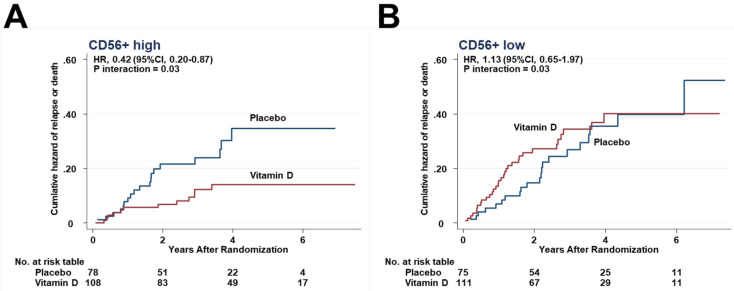
Effect of vitamin D on relapse or death in each subgroup. Nelson–Aalen cumulative hazard curves for relapse or death in the subgroups of (**A**) CD56+ higher half, and (**B**) CD56+ lower half.

**Figure 6 cancers-13-04708-f006:**
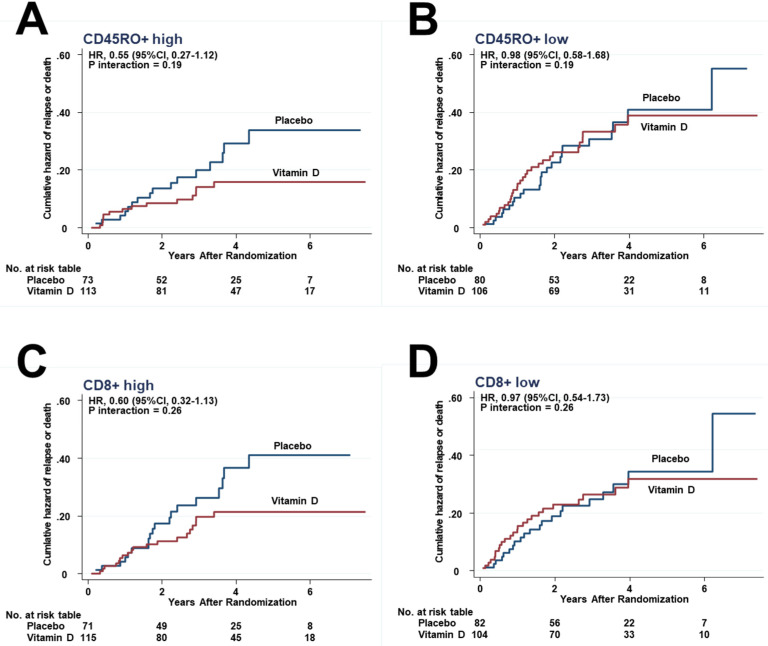
Effect of vitamin D on relapse or death in the subgroups of (**A**) CD45RO+ higher half, (**B**) CD45RO+ lower half, (**C**) CD8+ higher half, and (**D**) CD8+ lower half.

**Figure 7 cancers-13-04708-f007:**
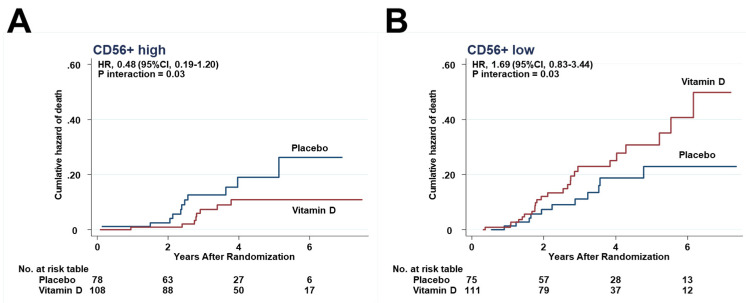
Effect of vitamin D on death in the subgroups of (**A**) CD56+ higher half, and (**B**) CD56+ lower half.

## Data Availability

The principal investigator had full access to all the data in the study and takes responsibility for the integrity of the data and the accuracy of the data analysis.

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
