# Peer review of "Effect of Vitamin D Supplements on Relapse of Digestive Tract Cancer with Tumor Stromal Immune Response: A Secondary Analysis of the AMATERASU Randomized Clinical Trial"

_cancers, 2021, doi:10.3390/cancers13184708_

Round 1

Reviewer 1 Report

Nowadays there is accumulating evidence concerning beneficial effects of vitamin D supplementation in variety of cancers and this interrelationship is under intense investigation. Experimental data show that vitamin D regulates the tumor microenvironment and is able to suppress growth of cancer cells. Dietary vitamin D supplementation was shown to modulate innate immunity by increasing natural killer activity in mice.

This original article presents a secondary subgroup-analysis of the AMATERASU trial in which a total of 372 patients were included. The participants  were divided into 2 subgroups stratified by the median density of each subset, i.e., CD3+, CD4+, CD8+, CD45RO+, and CD56+ immune cells, in tumor stroma into higher half (n=186) and lower half (n=186). Then, the effects of vitamin D supplementation on SHRs of relapse were compared between the higher and lower halves. In summary, Vitamin D supplementation significantly reduced the risk of relapse by 65% in the subgroup of patients with the higher half of CD56+ in tumor stroma, and did not change it in the lower half. The results and conclusion of this sub-analysis are of clinical importance and suggest a significant interaction between vitamin D supplementation and CD56+ cells. The authors have precisely described the limitations of the secondary analysis, but the later do not diminish the value of the scientific research data.

This original article is well structured and provides important knowledge about the role of Vitamin D in the process of tumorigenesis.

Author Response

Nowadays there is accumulating evidence concerning beneficial effects of vitamin D supplementation in variety of cancers and this interrelationship is under intense investigation. Experimental data show that vitamin D regulates the tumor microenvironment and is able to suppress growth of cancer cells. Dietary vitamin D supplementation was shown to modulate innate immunity by increasing natural killer activity in mice.

This original article presents a secondary subgroup-analysis of the AMATERASU trial in which a total of 372 patients were included. The participants were divided into 2 subgroups stratified by the median density of each subset, i.e., CD3+, CD4+, CD8+, CD45RO+, and CD56+ immune cells, in tumor stroma into higher half (n=186) and lower half (n=186). Then, the effects of vitamin D supplementation on SHRs of relapse were compared between the higher and lower halves. In summary, Vitamin D supplementation significantly reduced the risk of relapse by 65% in the subgroup of patients with the higher half of CD56+ in tumor stroma, and did not change it in the lower half. The results and conclusion of this sub-analysis are of clinical importance and suggest a significant interaction between vitamin D supplementation and CD56+ cells. The authors have precisely described the limitations of the secondary analysis, but the later do not diminish the value of the scientific research data.

This original article is well structured and provides important knowledge about the role of Vitamin D in the process of tumorigenesis.

Thank you. We appreciate your comments.

Reviewer 2 Report

General comments

The manuscript by Taisuke Akutsu et al. was done with samples previously used in two published papers: “Effects of vitamin D supplementation on relapse-free survival among patients with digestive tract cancers” and “Survival of digestive tract cancer patients with low bioavailable 25-hydroxyvitamin D levels”. The article is a well-written paper that addresses an important issue of interest to researchers in the field. Specifically, it studied how vitamin D affects the immune cell population in the tumors of supplemented patients and its relationship with relapse. There are, however, some comments that may help to improve the manuscript.

Major comments

  1. It would be very helpful for the readers if the authors included in the Introduction section a description of each immune cell type analyzed, as well as their significance and function in fighting cancer. Please describe the cell type expression of each antigen analyzed by IHC and if this depends on the stage of differentiation and state of activation of immune cells.
  2. Lines 186-187: How do authors explain the result showing that Ki-67 (a bad prognostic marker) was significantly higher in the CD3+ higher half than in the lower half? This bearing in mind that CD3+ T cells are considered predictive of better overall survival and indicative of an intratumor active immunologic state.
  3. Lines 25-26: “Vitamin D supplementation may reduce the risk of relapse in patients with a sufficient immune response in tumor stroma”. This phrase is confusing. Did vitamin D supplementation modify the immune response in tumor stroma? or just reduced the risk of relapse in patients who already had an adequate infiltration of immune cells in their tumors? Did vitamin D increase the number of CD56+ cells?
  4. What was the VD status in the patients with a sufficient immune response in tumor stroma, in comparison with the insufficient ones? Including a table with the final vitamin D serum levels in each group of patients would be very helpful.
  5. How was the CD56+ cells percentage with regards to vitamin D status? Was there a significant correlation between CD56+ density and VD serum status?
  6. Please provide IHC of VDR in the studied tissue samples, in order to observe the expression of this marker in tumor microenvironment (stroma) and cancer cells.
  7. The asseveration in line 308-309 seems to invalid the conclusions of the study, although the results are significant and very interesting. I don’t understand why almost half of the discussion is dedicated to highlight the limitations of the study, while very little is addressed at discussing the clinical significance of the results and their comparison with previous studies in the field. The way it is written gives the idea that the authors are not very confident of their results and conclusions. I suggest that, if they are certain of their findings, diminish the discussion of the limitations and increase the assessment of the clinical significance and possible applications in further studies or clinical translation.

Minor comments

  1. Line 21: Please change the word “accompanying”, it is not very clear in this context.
  2. Unfortunately, I could not have access to the supplementary data, which I believe was very important to the general assessment of the manuscript.
  3. Lines 198-199: Could you please explain more in-depth the meaning of the significant interaction found between vitamin D supplementation and the subgroup of CD56+ higher half?
  4. Please explain the difference between subdistribution hazard ratios and hazard ratios, and relate this to results in Figures 6 and 4.

Author Response

General comments

The manuscript by Taisuke Akutsu et al. was done with samples previously used in two published papers: “Effects of vitamin D supplementation on relapse-free survival among patients with digestive tract cancers” and “Survival of digestive tract cancer patients with low bioavailable 25-hydroxyvitamin D levels”. The article is a well-written paper that addresses an important issue of interest to researchers in the field. Specifically, it studied how vitamin D affects the immune cell population in the tumors of supplemented patients and its relationship with relapse. There are, however, some comments that may help to improve the manuscript.

Major comments

  1. It would be very helpful for the readers if the authors included in the Introduction section a description of each immune cell type analyzed, as well as their significance and function in fighting cancer. Please describe the cell type expression of each antigen analyzed by IHC and if this depends on the stage of differentiation and state of activation of immune cells.

Thank you for your suggestions. The function of each immune cell type was described as follows:

In fact, the meta-analyses of randomized clinical trials (RCTs) showed that vitamin D supplementation, compared with placebo, prevented the incidence of acute respiratory tract infection [4,5], probably through enhancing innate immunity, such as CD56+ natural killer cells. In addition, more tumoral immune cells have been shown to be associated with longer survival time of patients with various types of cancers, e.g., advanced ovarian cancer and CD3+ pan T cells [6], colorectal cancer and CD45RO+ memory T cells [7], and breast cancer and CD8+ cytotoxic T cells [8]. Moreover, in patients with head and neck cancer, a statistically significantly higher intratumoral and/or stromal infiltration of CD3+ pan T, CD4+ helper T, CD8+ cytotoxic T, CD56+ natural killer cells, and other immune cells was observed in the 25(OH)D-high patients compared with the 25-hydroxyvitamin D (25[OH]D)-low patients, which was associated with longer overall survival [9].

  1. Lines 186-187: How do authors explain the result showing that Ki-67 (a bad prognostic marker) was significantly higher in the CD3+ higher half than in the lower half? This bearing in mind that CD3+ T cells are considered predictive of better overall survival and indicative of an intratumor active immunologic state.

Thank you for raising this important point. We added a paragraph in the discussion as follows:

In the present study, Ki-67 was significantly higher in the CD3+ higher half than in the lower half. Ki-67 is considered to be an indicator of proliferative activity and thus one of the markers of a poor prognosis in carcinoid [28,29]. However, expression levels of Ki-67 were not always associated with survival of patients with cancer, at least in this cohort (data not shown). A recent study showed that high tumor cell proliferation assessed by Ki-67 was associated with increased levels of all tumor-infiltrating lymphocyte subsets, including CD3+ pan T cells in breast cancer [30], which is consistent with the results of the present study. Although the exact mechanisms are unknown, cancer cells with high proliferative activity may tend to induce inflammation and attract immune cells.

  1. Lines 25-26: “Vitamin D supplementation may reduce the risk of relapse in patients with a sufficient immune response in tumor stroma”. This phrase is confusing. Did vitamin D supplementation modify the immune response in tumor stroma? or just reduced the risk of relapse in patients who already had an adequate infiltration of immune cells in their tumors? Did vitamin D increase the number of CD56+ cells?

Thank you for pointing this out. We modified the text as follows:

Simple Summary

Line 25~26

Vitamin D supplementation may reduce the risk of relapse in patients who already had an adequate infiltration of immune cells in their tumor stroma.

Vitamin D supplementation may reduce the risk of relapse in patients who already had adequate infiltration of immune response cells in their tumor stroma.

Abstract

Line 39-40

In patients with digestive tract cancer, vitamin D supplementation was hypothesized to reduce the risk of relapse in the subgroup of patients who already have an adequate infiltration of immune cells in their tumor stroma.

Conclusions

Line 344-346

Vitamin D supplementation may reduce the risk of relapse in the subgroup of patients with digestive tract cancer who already had an adequate infiltration of immune cells in their tumor stroma.

  1. What was the VD status in the patients with a sufficient immune response in tumor stroma, in comparison with the insufficient ones? Including a table with the final vitamin D serum levels in each group of patients would be very helpful.

Thank you for raising this important point. Please refer the following sentence and Table S2:

Line 195-198

Patients’ characteristics in the subgroups are shown in Table S2. There were no differences in serum 25(OH)D levels or bioavailable 25(OH)D levels before vitamin D intervention between the compared subgroups, e.g., patients with between CD56+ cells high and low.

  1. How was the CD56+ cells percentage with regards to vitamin D status? Was there a significant correlation between CD56+ density and VD serum status?

Thank you for this question. There was no significant correlation between CD56+ density and VD serum status. We added the following explanation:

Results

Line 185-187

There were no differences in serum 25(OH)D levels or bioavailable 25(OH)D levels before vitamin D intervention between the compared subgroups, e.g., patients with between CD56+ cells high and low.

  1. Please provide IHC of VDR in the studied tissue samples, in order to observe the expression of this marker in tumor microenvironment (stroma) and cancer cells.

Thank you for this suggestion. We added Figure 2 showing VDR in order to show the expression of this marker in the tumor microenvironment (stroma) and cancer cells.

  1. The asseveration in line 308-309 seems to invalid the conclusions of the study, although the results are significant and very interesting. I don’t understand why almost half of the discussion is dedicated to highlight the limitations of the study, while very little is addressed at discussing the clinical significance of the results and their comparison with previous studies in the field. The way it is written gives the idea that the authors are not very confident of their results and conclusions. I suggest that, if they are certain of their findings, diminish the discussion of the limitations and increase the assessment of the clinical significance and possible applications in further studies or clinical translation.

Thank you for your encouraging comments. We shortened the discussion of the limitations and increased the assessment of the clinical significance and possible applications in further studies or clinical translation.

Minor comments

  1. Line 21: Please change the word “accompanying”, it is not very clear in this context.

Thank you for suggestion. We changed it as follows:

However, interactions among vitamin D supplementation, intratumoral immune cells, and cancer relapse have not yet been elucidated. The aim was to assess the effects of vitamin D supplementation on relapse in patients with digestive tract cancer with immune response in tumor.

  1. Unfortunately, I could not have access to the supplementary data, which I believe was very important to the general assessment of the manuscript.

Please refer to the supplementary data.

  1. Lines 198-199: Could you please explain more in-depth the meaning of the significant interaction found between vitamin D supplementation and the subgroup of CD56+ higher half?

We have provided a more in-depth explanation, as follows:

Results

Line 211-212

There was a significant interaction between vitamin D supplementation and the subgroup of CD56+ higher half (Pinteraction = 0.02). In other words, vitamin D supplementation effectively reduced the risk of relapse only in the subgroup of CD56+ cells higher half, but did not alter the risk in its lower half.

  1. Please explain the difference between subdistribution hazard ratios and hazard ratios, and relate this to results in Figures 6 and 4.

Thank you for pointing out. We changed it as follows:

Methods

Statistical analysis

Line 157-164

To evaluate the effects of vitamin D supplementation on relapse as the primary outcome, cumulative incidence functions were applied by considering patient deaths due to causes other than cancer relapse as a competing risk; competing risk regression was performed using subdistribution hazard ratios (SHRs) and 95% confidence intervals (95%CIs) [17]. On the other hand, because the competing risk between relapse and death do not have to be considered, the effects of vitamin D and placebo on the risk of relapse or all-cause death as well as the all-cause death were estimated using Nelson–Aalen cumulative hazard curves for outcomes.

Results

Line 203-205

Considering competing risk between relapse and death, the effects of vitamin D supplementation on relapse were compared between the higher and lower halves of each immune subset in tumor stroma by SHR.

Round 2

Reviewer 2 Report

The authors have adequately responded to the comments. A minor observation is that Figure 2 is not referred in the manuscript text.